# Effects of Exercise Programs on Psychoemotional and Quality-of-Life Factors in Adult Patients with Cancer and Hematopoietic Stem Cell Transplantation or Bone Marrow Transplantation: A Systematic Review

**DOI:** 10.3390/ijerph192315896

**Published:** 2022-11-29

**Authors:** Erica Morales Rodríguez, Jorge Lorenzo Calvo, Miriam Granado-Peinado, Txomin Pérez-Bilbao, Alejandro F. San Juan

**Affiliations:** 1Sports Department, Faculty of Physical Activity and Sports Sciences-INEF, Universidad Politécnica de Madrid, 28040 Madrid, Spain; 2Department of Health and Human Performance, Faculty of Physical Activity and Sports Sciences-INEF, Universidad Politécnica de Madrid, 28040 Madrid, Spain; 3Faculty of Education and Psychology, Universidad Francisco de Vitoria, 28223 Madrid, Spain

**Keywords:** cancer, exercise, hematopoietic stem cell transplantation, psychoemotional factors, quality of life

## Abstract

This review analyzed the effects of an exercise program on psychoemotional and quality-of-life (QoL) factors in adult patients with cancer and hematopoietic stem cell transplantation (HSCT) or bone marrow transplantation (BMT). Studies were identified from the PubMed and Web of Science databases (from inception to 24 August 2022), according to the Preferred Reporting Items for Systematic Reviews and Meta-Analyses (PRISMA) checklist. The methodological quality of the included studies was assessed with the Physiotherapy Evidence Database (PEDro) scale, based in turn on the Delphi list. A total of 20 randomized controlled studies were included with 1219 participants. The main result of this systematic review is that exercise program interventions produce improvements on psychoemotional and QoL factors in adult patients with cancer and HSCT or BMT. Moreover, exercise programs may have a beneficial effect on health, maintaining or increasing the patient’s QoL. Further, it has a positive effect on the prevention and control of transplant complications in combination with medical treatment.

## 1. Introduction

Bone marrow transplantation (BMT) or hematopoietic stem cell transplantation (HSCT) is an effective treatment for adults with hematological diseases [1]. This therapy consists of total (myeloablative) or partial (nonmyeloablative) ablation of the bone marrow with chemotherapy and/or radiotherapy, followed by the administration of HSCT. The main aim of this therapy is the reconstitution of the hemopoietic system by transferring the pluripotent cells present in the bone marrow (stem cells) [2]. There are two possibilities of HSCT: (1) Allogeneic HSCT uses blood and marrow from similar donors, and it may be associated with graft versus host disease [3,4]. Graft versus host disease is caused when the donor’s cells react against those of the immunosuppressed receptor. Long-term graft versus host disease is a prognostic factor for overall survival [5], with a 59% probability of developing chronic cardiovascular, endocrine, gastrointestinal, or possibly renal disease with different degrees of severity (e.g., stroke, diabetes, cirrhosis) [6]. (2) In autologous HSCT, the stem cells are harvested from the individual’s bone marrow or blood for reinfusion after prior antineoplastic conditioning therapy [7,8].

After oncological treatment, many patients develop long-term psychoemotional disorders that reduce their quality of life (QoL) [9], as fatigue [10,11,12], emotional state difficulties resulting from anxiety and depression [9], distress [13], or sleep habits [14]. These psychological disturbances are often due to the emotional stress of the disease, the treatment, the disease progression, the fear of recurrence, or the lack of information or resources to manage the disease [15]. Moreover, HSCT could cause alterations in human body systems [16] with reduced exercise capacity and skeletal muscle oxygenation [17].

Of the variables mentioned above, (a) *cancer-related fatigue* is one of the most common and disabling symptoms experienced by cancer patients [18]. In the context of cancer, cancer-related fatigue is a “distressing, persistent, subjective sense of physical, emotional or cognitive tiredness or exhaustion related to cancer or cancer treatment that is not proportional to recent activity and interferes with usual functioning” [19]. Thus, fatigue is also associated with psychological and psychosocial disorders [12]. Cancer-related fatigue must be distinguished from other types of physical fatigue, which may improve with rest alone [20], or from other forms, such as chronic fatigue syndrome, which, although they may have similar psychological manifestations and symptoms, respond to different pathophysiological mechanisms [21]. (b) *Depression and anxiety* emerge in the cancer patient as a result of reaction to diagnosis, therapy, or relapse or during survivorship. It is pathological if it compromises the person’s normal functioning and if it is disproportionate to the level of threat [22]. Depression and anxiety are common in some types of cancer patients and are associated with a lower QoL [23]. (c) *Psychological distress* is considered a serious problem for patients diagnosed with cancer. It is a “multifactorial unpleasant emotional experience of a psycho-social, and/or spiritual nature that may interfere with the ability to cope effectively with cancer, its physical symptoms, and its treatment” [24]. Cancer patients in more advanced stages, with worse prognosis and higher disease burden and perhaps those who are younger, seem to be at the highest risk of psychological distress [25,26]. (d) *Sleep habit* influences cognition, attention, memory, and learning. Moreover, according to cross-sectional studies in cancer patients, poor sleep quality is associated with poor cognitive functioning [27]. (e) *QoL* evaluates the person‘s well-being in relation to diverse emotional, social, and physical dimensions of life, which may be influenced by disease or disability. The measurement of QoL can contribute to modify therapies and to prevent secondary effects or diseases [28].

Until 1960, the recommendation of exercise was only reported for cardiac, pulmonary, and renal physical pathologies. In 1989, significant improvements in maximal oxygen uptake (VO2max) were found in 45 women with breast cancer after postchemotherapy with 10 weeks of aerobic exercise intervention [29]. In 1996, a pioneering study in cancer patients after BMT demonstrated significant improvements in VO2max with an aerobic exercise program to recover functional capacity for 6 weeks [30].

However, from 1989, a first quasi-experimental study reported a home exercise program that improved the performance of 12 leukemia patients with 30 min and 3 sessions per week at 85% of heart rate maximal. No significant results were obtained in functional capacity or in the reduction of depression by Beck’s scale [31]. Subsequently, in 1994, a study significantly improved the mood of nine women with breast tumor during chemotherapy in an unstructured walking program [32]. In 1997, 32 lymphoma patients undergoing a supervised treadmill walking program for 6 weeks reported less fatigue through an interview in the experimental group [33]. In 1999, a systematic review demonstrated that physical exercise had a positive effect on QoL in oncology patients, including physical and functional well-being [13].

Specifically, exercise programs in HSCT-BMT patients are now known to produce positive effects in QoL, which is a predictor of physical, functional, and emotional well-being [34,35,36,37]. Further, exercise programs improve also mental health [38,39] with reduction in perceived fatigue, anxiety, and depression [1,40]. Nevertheless, the existing studies have limitations, for example, the joint inclusion of allogeneic and autologous patients [30], few control groups or a small sample size [41], disproportion sample between the control group and the intervention group [42], and few days of monitoring the activity program [43].

Our outcomes are in accordance with two systematic reviews and meta-analyses [44,45], which examined the effects of exercise on psychoemotional and QoL factors in adult patients with cancer and HSCT or BMT.

Then, the purpose of this manuscript is to systematically review the current literature and analyze the scientific evidence that have examined the effects of exercise programs on psychoemotional and QoL factors in adult patients with cancer and HSCT.

## 2. Materials and Methods

### 2.1. Systematic Search

The systematic review is reported in accordance with the Preferred Reporting Items for Systematic Reviews and Meta-analyses (PRISMA), as well as possible consequences for the risk of bias [46]. An electronic search of papers produced in English was realized in the databases PubMed and Web of Sciences (from the inception until 24 August 2022). The full search strategy for PubMed with the MeSH-indexed terms was: (bone marrow transplant OR hematopoietic stem cell transplantation) AND (exercise OR physical activity) AND (acceleration OR anxiety OR cancer survivors OR cardiovascular OR fatigue OR health OR immune system OR infection OR mobility OR neoplasms OR neuromuscular function OR oxygen consumption OR quality of life OR range of motion OR resistance training OR strength OR stress). Moreover, we realized the review of the reference lists of relevant systematic reviews and meta-analysis and backward citation of included studies to ensure additional potential studies that could have been missed through database searches.

### 2.2. Selection of the Studies

Studies meeting the following criteria were included in this review: (a) published studies, (b) published in English, (c) randomized controlled trials (RCTs), (d) adult patients (age ≥ 18 years old) of both sexes who suffered or had suffered from any type of cancer at the time of the study, (e) patients in the process of receiving or who received an HSCT, (f) patients who had undergone an exercise program intervention, (g) studies with an effect on psychoemotional and QoL factors in adult patients, and (h) only studies with a score ≥6 in the Physiotherapy Evidence Database (PEDro) scale [47,48,49,50,51]. Studies that were excluded in this review were: (a) unpublished clinical trials registered on clinicals.gov, (b) grey literature (e.g., reports, conference proceedings, doctoral theses), (c) published and unpublished systematic reviews or meta-analyses, and (d) studies that only related exercise and drugs.

The study selection process was conducted via step 1 (title and abstract screening), followed by step 2 (full-text screening). In step 2, the studies were evaluated with the PEDro scale to select those that scored ≥ 6 points.

### 2.3. Data Extraction

The following variables from the included studies were extracted independently by two authors (E.M.R. and J.L.C.): sample (N), sex and age of participants, type of cancer, characteristics of the interventions (type, equipment, frequency, intensity, duration, session, rest, supervision, adjustment), and effects of psychoemotional and QoL variables. At the same time, discussion and consensus on data extraction resolved discrepancies. The exercise programs before and after HSCT in cancer patients were separated.

### 2.4. Risk of Bias Assessment

Two authors (E.M.R and J.L.C.) independently scored the studies using the PEDro scale, based, in turn, on the Delphi list [52], and disagreements were resolved through discussion with a third author (T.P.B.). The total score of the PEDro scale was from 0 to 10, counting the number of criteria met by each study (see footnotes in Table 1). The quality of the study was rated as poor (PEDro score ≤ 3), fair (4–5), or high (≥6). We excluded studies with PEDro < 6 using the criteria described in the index in Table 1. Studies with a PEDro score < 6 were excluded in step 2 (full-text screening).

## 3. Results

### 3.1. Study Selection

A total of 1321 references were identified in the database. After removing duplicate studies (*n* = 646) and removing studies for other reasons (i.e., patients under 18 years old, language, RCTs, human) (*n* = 453), 222 studies were included. After screening by title and abstract (*n* = 150) and full text (*n* = 72), 202 studies were excluded, which were excluded by PEDro (*n* = 14), and 20 studies overcame all the inclusion criteria (Figure 1).

### 3.2. Study Characteristics

A total of 20 studies were included and reviewed in this systematic review [34,40,41,42,43,53,54,55,56,57,58,59,60,61,62,63,64,65,66,67] (Table 1). The characteristics of the included studies are shown in Table 2 and Table 3. All studies included were RCTs with a control group (CT) with no exercise intervention and an intervention group (EXP), except the study [62], which divided the sample size into a supervised or self-directed exercise program.

The work performed by the CGs in the different studies consisted of: in 2 studies (10%), they performed mobilization and/or stretching exercises [32,57]; in 3 studies (15%), they received advice that moderate physical activity is favorable during the transplant process [38,55,56]; in 4 studies (20%), patients received physiotherapy treatment [42,50,52,54]; in 5 studies (25%) patients received recommendations regarding physical activity but were not restricted in their physical activities [40,46,48,58,59]; in 1 study (5%), patients received false inspiratory muscle training; in 1 study (5%), patients performed a home-based exercise program that was not described [45]; and in 1 study (5%), a home-based exercise program consisting of strength training with elastic bands in addition to walking was performed. Strength training was performed 3 times/week, 1–3 sets, 10–15 repetitions in 10 whole-body exercises. Walking training was performed 3 times/week with progressive duration up to 30 min/session [47]; in 3 studies (15%) the tasks performed by the patients were not described [41,51,53].

### 3.3. Quality Assessment and Publication Bias

The quality of the 20 studies included was high (median PEDro score = 6, range = 6–10; Table 1). There were 2 studies with a score = 6 (10%), 15 studies with a score = 7–8 (75%), and 3 studies with a score = 9–10 (15%). The complete results can be found in Table 1.

### 3.4. Characteristics of Participants

The analyzed studies included a total of 1219 oncologic patients (age range = 18 to 75 years). They were divided into an EXP group with 606 patients (230 women) and a CT group with 613 patients (302 women), but there were some studies [41,64] that did not account for the number of females in the experimental or control group and, therefore, could include more females. The main cancers in the patients were leukemia (80%), lymphoma (60%), myeloma (75%), and myelodysplasia (55%).

### 3.5. Characteristics of Exercise Interventions

The characteristics of the exercise interventions (Table 2 and Table 3) were very heterogeneous. In total, 11 studies (55%) analyzed the effects of an exercise program before and after HSCT treatment [34,40,42,56,57,59,63,64,65,66,67], and 9 studies (45%) analyzed the effects after HSCT [41,43,53,54,55,58,60,61,62].

*Supervision:* Exercise programs were supervised by investigators in 15 studies (75%) [34,41,42,43,53,54,55,57,58,59,60,61,62,63,65], and semisupervised in 3 studies (15%) [40,56,66], and 2 studies (10%) were not supervised [64,67].

Frequency: A total of 3 studies (15%) applied the exercise intervention 7 days per week [55,56,67], 9 studies (45%) administered the intervention 5 days per week [34,40,42,43,57,58,59,63,65], 4 studies (20%) 3–4 days per week [41,60,62,66], 1 study (5%) 2 days per week [53], and 2 studies (10%) 1–2 days per week [54,61]. Only 1 study (5%) did not report the frequency of exercise [64].

Intensity training: A total of 7 studies (35%) did not provide information for the type of intensity training [56,57,58,59,60,64,67]. A total of 2 studies (10%) did not provide any details regarding strength intensity training [53,62], and 1 study (5%) did report information regarding aerobic training [43]. In 6 studies (30%), the exercise intensity was estimated using a rate of perceived exertion scale [40,41,42,64,65,66], 5 studies (25%) reported aerobic exercise intensity using a percentage of maximal heart rate [34,42,53,62,65], 3 studies (15%) reported resistance exercise intensity using a percentage of one maximum repetition [43,54,61], 3 studies (15%) reported aerobic exercise intensity with a percentage of power [54,61,63], 1 study (5%) informed of the exercise intensity using a percentage of maximal inspiratory pressure [55], and 1 study (5%) informed of the exercise intensity with an increase in the steps to realize by 10% weekly [56].

*Duration:* A total 11 studies (55%) were up to 8 weeks [34,41,42,43,55,56,59,62,63,65,66]; 6 studies (30%) were between 9 and 18 weeks [40,53,54,60,61,67]; 1 study (5%) was over 18 weeks [64]; and in 2 studies (10%), this information was not completely reported [57,58]. In 6 studies (30%), the duration of each session varied between 20 and 40 min [34,55,58,59,63,66]; in 5 studies (25%), between 15 and 60 min [42,57,61,65,67], and in 9 studies (45%), the duration was not correctly reported [40,41,43,53,54,56,60,62,64].

*Type:* The majority of the studies included multicomponent exercise interventions. A total of 9 studies (45%) were focused on strength and aerobic training [40,43,53,54,56,59,61,62,64]; 2 studies (10%) combined strength, aerobic training, stretching, and relaxation [42,65]; 2 studies (10%) incorporated aerobic, stretching, and activities of daily living training [34,63]; 1 study (5%) included aerobic training [67]; 1 study (5%) included strength training [41]; 1 study (5%) combined strength training plus whole-body vibration (WBV) [57]; 1 study (5%) included aerobic training and relaxation [60]; 1 study (5%) included aerobic training, strength, and stretching [58]; 1 study (5%) included aerobic training and respiratory muscle [55]; and 1 study (5%) included aerobic training and activities of daily living [66]. When reporting, the most frequently used activities for aerobic training were walking or running on a treadmill (60%) and cycling (40%). For strength training, free weight (40%), a dynamometer (30%), or elastic bands (20%) were typically used. For respiratory muscle exercises, pressure threshold loading devices (5%) were used. For strength training with whole body vibration, a vibration platform (5%) was used. For a more detailed description, see Table 2 and Table 3.

*Delivery setting:* A total of 10 interventions (50%) were conducted in healthcare settings (e.g., hospitals) [34,42,43,54,57,58,61,63,65,66], 8 interventions (40%) were conducted in both participants’ homes and healthcare settings [40,41,55,56,59,60,62,67], 1 intervention (5%) was carried out in the participant’s home [64], and 1 intervention (5%) was implemented in both the physiotherapy office and the gym [53].

### 3.6. Compliance Rate

A total of 10 of the 20 papers (50%) presented the compliance rates of their subjects with exercise, with the adherence data for the patients shown as follows: <40% [62,67], 40–80% [41,57,64,66], 80–90% [54,65], and >90% [40,56]. For a detailed description of the studies, see Table 2 and Table 3.

### 3.7. Endpoints and Exercise Intervention Psychoemotional Results

#### 3.7.1. Psychoemotional Variables

Psychoemotional variables were examined in 14 of the 20 studies (70%). The next variables of the fatigue factor were directly monitored using a Fatigue Impact Scale (5%) [55], Brief Fatigue Inventory Scale (5%) [62], Multidimensional Fatigue Inventory (56%) [40,43,54,57,59,61], and Chalder fatigue subscale (5%) [56]. Other psychoemotional variables evaluated were the state of mind with: the Profile of Mood States (10%) [40,64] and the Scale for Mood States Assessment (5%) [67]. In addition, anxiety and depression were assessed with: the Hospital Anxiety and Depression Scale (44%) [40,42,54,58], subscale of the Montgomery–Asberg Depression Rating Scale (5%) [55], Distress Thermometer (33%) [40,58,59], and PROMIS (5%) [56] with the measurement of anger, anxiety, and depression. Finally, health status was assessed using the 36-Item Short Form Survey (10%) [58,60].

In 7 of the 14 the studies (50%), the results suggested a significant increase in the following psychoemotional variables: (a) fatigue measures with the scales: Fatigue Impact Scale [55], Multidimensional Fatigue Inventory [40,57], and Chalder fatigue subscale [56]; (b) anxiety and depression measures with the scales: Hospital Anxiety and Depression Scale [58] and Montgomery–Asberg Depression Rating Scale [55]; (c) mood state measures with the scales: the Profile of Mood States [40,64] and the Scale for Mood States Assessment [67]; (d) health status with the scale: 36-Item Short Form Survey [58]. Only 2 of the 7 studies [55,56] showed improvement in psychoemotional variables with an exercise program applied only after HSCT.

Participants in the studies in which these variables were increased had: (a) a mean age between 49 and 58 years, (b) 38.8%–41.7% were women in the experimental group, and (c) the primary cancer in these studies was multiple myeloma, although no patient’s cancer stage was detailed, which does not allow us to find out whether the improvement is the same for all patients or not.

#### 3.7.2. Quality of Life Variables

QoL variables were examined in 19 of the 20 studies (95%). The variable QoL was measured through the different scales: (a) QoL measures with the scales: Symptom Scales of the EORTC Myeloma Module (5%) [54], QLQ-C30 (45%) [34,40,42,43,48,49,50,51,52,54,58,60,63], quality of life scale (5%) [57,60,66], quality of life index [68], Functional Assessment of Cancer Therapy-Bone Marrow Transplant (14%) [42,53,58,66], General Health Questionnaire (5%) [60]; (b) QoL to assess the ability of cancer patients to perform daily oncological tasks measure with the scale: Karnofsky Performance Status (10%) [42,67], and (c) the sleep was assessed with actigraphy with parameters: total minutes of nighttime sleep [64], total sleep time [56], duration of awakenings after sleep onset, and the number of awakenings [56]. Additionally, sleep was assessed with the daytime sleepiness with the ESE (5%) [64], the sleep disturbance subscale of QLQ-C30 (5%), and the sleep disturbance subscale of PROMIS (5%) [56].

In 10 of 19 studies (53%), the results suggest a significant increase in the following QoL variables before and after HSCT: (a) QoL measures with the scales: QLQ-C30 [43,50,51,52,58], health-related quality of life [48], quality of life [66], Functional Assessment of Cancer Therapy-Bone Marrow Transplant [58,66]; (b) ability of cancer patients to perform daily oncological activities, measured with the scale: Karnofsky Performance Status [67]; and (c) sleep measure with the parameters: total minutes of nighttime sleep [59]. Only 4 of the 9 studies [43,50,53] showed improvement in psychological quality of life variables with an exercise program applied after HSCT.

Patients in the studies in which these variables were increased shared: (a) a mean age between 34.1–62.2, (b) 16–47% were women in the experimental group, and (c) the common cancers in these studies were leukemia and multiple myeloma, without specifying the stage of the cancer, not being able to establish whether the improvement is the same for all patients.

## 4. Discussion

This systematic review of the literature analyzed the effects of the exercise programs on psychoemotional and QoL factors in adult patients with cancer and HSCT-BMT. Only randomized controlled trials were included, providing a high selection of methodologically sound studies. The main finding of this study is that exercise program interventions produced significant improvements in psychoemotional and QoL variables of over 50% in those studies analyzed before and after HSCT, being less than 40% in studies analyzed exclusively after HSCT.

Our results are in line with two systematic reviews and meta-analyses that examined the effects of exercise on psychoemotional and QoL factors in adult patients with cancer and HSCT or BMT. The review dates were, respectively: from 1 January 2010 to 1 July 2020 and from inception to 5 December 2020. Our systematic review presents similarities with these reviews: these two reviews included 7 of the 20 studies that are also discussed in our review [44,45], adherence to exercise was recorded, global QoL and fatigue were studied, and exercise interventions were heterogeneous. However, there are some relevant differences between the present systematic review and these studies [44,45]: The present review included only RCTs, and it compared a higher number of RCTs only with an exercise program intervention (20 versus 11 [44], 19 [45] studies, respectively). It was focused on specific QoL (i.e., symptom scales, oncological task capacity, habit of sleep) and psychoemotional factors (i.e., fatigue, mood state, anxiety and depression, health status), and not only in global QoL [44], and global QoL and fatigue [44,45]. Moreover, it excluded studies with nutrition intervention to prevent potential intervening variable, not another systematic review [44]. Further, a recent meta-analysis in patients with hematologic cancer published in 2022, although in another clinical situation (i.e., without HSCT transplantation), observed the same lines of positive effects after an exercise intervention [69]. Therefore, the present systematic review has added to the knowledge of studies a higher methodological quality, observing improvements in specific QoL factors and in psychoemotional factors, specifically in fatigue, anxiety and depression, mood, and health.

### 4.1. Psychoemotional Variables

A total of 8 of 14 studies (57%) observed improvements in psychoemotional variables: fatigue, mood state, anxiety and depression, and health status. In the case of fatigue, the exercise interventions showed a mean improvement of 16.1% (6.5–34.1%) in the different dimensions used in the tools used to measure fatigue [40,50,51,52]. In the Multidimensional Fatigue Inventory, the mean improvement in the general fatigue dimension was 25.4% [40,52], while the improvement in the physical fatigue dimension was 8.1% [40]. In this sense, only one study [40] established correlations between the variables analyzed. It reported that the higher the physical capacity, the significantly lower the fatigue levels of the patients. Moreover, the Fatigue Impact Scale was used in a study [50] where an improvement of 24.1% was found, and finally, the Chalder fatigue subscale was used in another study [51] whose mental fatigue dimension improved by 6.5% in the experimental group, but it should be noted that the control group improved more by 12.9%. We would also like to point out that the Profile of Mood States scale has a dimension on fatigue that showed an improvement of 15.2% [40].

Regarding the state of mind variable, two studies [40,59] used the Profile of Mood State scale, showing an improvement in the total mood disturbance score [59]; unfortunately this study did not present the raw data, so the percentages of improvement cannot be calculated. Another study that used the Profile of Mood State scale [40] did not include the vigor–activity score dimension, and therefore, it is not possible to calculate the total mood disturbance score. However, the anger/hostility dimension showed an improvement of 36.0% and 20.0% at discharge and 6–8 weeks after discharge, respectively [40]. In addition, the depression dimension measured by this scale showed a decrease of 3.5% from discharge to 6–8 weeks later in the experimental group, while the control group worsened by 33.1% in the same period [40]. The state of mind was also assessed using the emotional well-being questionnaire/Scale for Mood States Assessment [62]. Unfortunately, as has been the case in other studies, the raw data were not reported so that percentages of improvement cannot be established.

The depression variable has also been measured using the Montgomery–Asberg Depression Rating Scale, which also takes anxiety into account [50], in which the experimental group showed a 53.1% improvement in depression, and the Hospital Anxiety and Depression Scale [53], which showed a 30.9% improvement in depression in the experimental group. Anxiety did not present raw data to calculate percentages, and the study indicates that in the experimental group, anxiety levels decreased, while those of the control group remained unchanged.

Finally, the health status variable showed an improvement in the experimental group of 18.2% in the vitality domain of the 36-Item Short Form Survey.

The Multidimensional Fatigue Inventory is the main tool to assess fatigue globally [70], and the Profile of Mood State is psychology’s primary instrument for assessing six dimensions of the mood construct: anger, confusion, depression, fatigue, tension, and vigor [71]. Fatigue and symptomatic distress are associated with psychological disorders, resulting in a decrease in functional status [12]. Thus, the exercise program has an important protective function for the psychoemotional aspect of the patient during the most intensive periods of the treatment [72], and can even improve fatigue parameters, according to Abo et al. [73]. Exercise reduces cancer-related fatigue and increases oxygen transport from the blood and its use by tissues, and psychologically moderates mood and sleep disorders, depression, and anxiety [74]. Therefore, the benefits are better if an exercise program is used in combination with medical treatment [75]. Our data do not correspond with those of other reviews in terms of the fatigue measurement scales used: they showed no significant data on the Multidimensional Fatigue Inventory and Functional Assessment of Cancer Therapy-Bone Marrow Transplant, although they did show a decrease in the QLQ C-30 fatigue subscale [44], and showed a decrease in the QLQ C-30 fatigue subscale and did not use other fatigue measurement tools [45]. Although a variety of fatigue measurement tools, including Brief Fatigue Inventory Scale, were used, there were no significant results [69].

On the other side, 6 of the 14 studies (43%) found no significant improvement after the exercise interventions. This absence of positive effect may be due to the following reasons: the patient sample consisted exclusively of allogeneic transplant recipients [62]; the treatment combined with autologous stem cell transplantation for solid tumors [43]; there was low sample size [42,43,60,62]; the variety of exercise types in each group varied the volume of exercise between supervised and self-directed groups [62]; overall levels of daily physical activity were not quantified [62]; few counselling sessions made long-term follow-up of changes in daily physical activity behavior difficult [61]; the QLQ C-30 was used instead of the Multidimensional Fatigue Inventory, which focuses especially on the complexity of fatigue dimensions and psychosocial parameters were not assessed [43]; poor patient attendance made it difficult for patients to receive information, reduced their control, and increased their psychological distress [60]; and patients, therapists, and outcome evaluators were not blinded, and in the CT, there was a higher percentage of female patients [63].

According to the studies analyzed, the most appropriate exercise program to improve psychoemotional variables had the following characteristics: it was supervised, with a duration of 12 weeks and sessions of between 20 and 40 min. It combined strength and aerobic training and was performed in health facilities. Unfortunately, almost all the studies did not adequately report the intensity of training, so the most appropriate intensity cannot be proposed.

### 4.2. Quality of Life Variables

In 10 of 19 studies (53%), improvements were observed in the measured of QoL variables. Regarding the QoL variable, the experimental group showed improvements in the following dimensions: the physical functioning dimension showed a mean improvement of 4.9% (3.3–6.5%) [43,50]; the emotional functioning dimension improved on average by 5.8% (3.4–8.2%) [48,51]; the global health status dimension improved on average by 13.9% (12.6–15.1%) [50,61]; the physical well-being dimension improved on average by 30.5% (21.1–39.8%) [53,61]; the emotional well-being and functional well-being dimensions improved by 4.75% [53] and 27.5% [61], respectively; the emotional state dimension improved by 26.9% [58]; and the hematopoietic stem cell transplantation-related QoL dimension improved by 13.5% [61]. In the specific case of the Karnofsky Performance Status [62], the experimental group worsened less than the control group, with a decrease of 10 points compared with the 20 points lost by the control group (KPS scale ranges from 0 to 100 points). In one study [52], the improvement of the experimental group was found in the follow-up period, where the dimensions physical functioning, role functioning, emotional functioning, and social functioning improved by 8.4%, 33.4%, 35.8%, and 33.4%, respectively. Finally, in the case of sleep, a significant improvement was found in total minutes of nighttime sleep [59]. Unfortunately, this study did not present the raw data so that the percentages of improvement could not be calculated.

The most used QoL tool in oncology is QLQ-C30 [76], which also is the recommended version for all clinical trials [77], and the Functional Assessment of Cancer Therapy-Bone Marrow Transplant, which contains an HSCT-specific scale and is designed to measure five dimensions of the QoL [78].

Studies indicate that physical exercise after cancer diagnosis has a positive effect on QoL in emotional well-being [13]. This was reported by one study of exercise intervention after HSCT, in which patients showed an improvement in global QoL [35]. Therefore, exercise appears to have a healthy effect on health before and after transplantation, maintaining or even increasing the individual’s QoL. Our data showed parallels with other reviews in terms of QoL measurement: showed significant QoL improvement data on the QLQ C-30 global QoL scale [44], and showed significant QoL improvement data on the health-related quality of life, although they did not use the QLQ C-30 scale [45] and used, among others, QLQ C-30, Karnofsky Performance Status, and the 36-Item Short Form Survey, significantly increasing the quality of life in the combined exercise group [69].

However, 12 of the 19 studies (63%) found no significant improvements after the exercise interventions. This lack of improvement may be caused by the following reasons: no randomization of baseline fitness characteristics prior to exercise intervention, independently of the severity of pathology or symptomatology [59]; a small sample size [41,51,52,60]; strict exclusion criteria, limitations of time to access the program and dropout due to side effects of treatment [42]; no significant results, as immediate exercise after transplantation does not necessarily accelerate natural recovery from disease [61]; a number of exercise data records before and during hospitalization were missing [40]; documentation of social support in training was not completed, impacting adherence to training [40]; CT was an active group and used a pedometer [40]; two CT patients exercised 4 weeks after hospital discharge, which created a confusion effect [41]; treatment-related complications coupled with a slow recovery process [65]; it was not a blind study, and the EXP received 18% more physical activity [34]; contamination in the CT in relation to the physiotherapy care received [54]; and the experimental group and control group performed the same amount of physical activity [48].

Exercise program interventions appear to be safe in adult oncology transplant patients and produce benefits in cardiorespiratory fitness, endurance, functional capacity [50], and quality of life [79]. It is beneficial to start before or just after transplantation [80]. Finally, it has an important role in mental performance by decreasing anxiety, depression, or fatigue [81].

As a practical application of the present systematic review, the program that produced improvements in QoL was supervised, with a frequency of 5 days a week, a duration of 8 weeks, and sessions with duration between 15 and 40 min. It combined strength and aerobic training, and was carried out in health facilities. Again, intensity was not well described in most studies, but of the few studies that did describe it, it appears that the most usual intensity of strength and aerobic training was moderately low.

## 5. Limitations

This review reveals that the literature published so far on this theme has limitations in respect to: (a) the small sample size (13 of 19 studies) including <100 participants; however, it must consider that sometimes it may be hard to collect data from numerous patients with the same cancer type and treatment and to meet all the inclusion and exclusion criteria; (b) heterogeneous characteristics of the sample (e.g., wide age range, different types of cancer and time since diagnosis and end of treatment); (c) heterogeneity of exercise interventions (e.g., different types, frequency, intensity, type of session, volume of session, type of supervision); (d) lack of follow-up [57,67]; (e) control of the compliance rate is not specified [82] or is less than 40% [62,67]; (f) the evaluators are not blinded to group assignment [56]; and (g) the stage of cancer of each patient is not reflected in any of the articles analyzed.

Future lines of research need to center on RCTs in progressing towards new exercise programs with different load planning and homogeneity and long-term follow-up of the exercise program to blind the evaluators to group assignment and description of the cancer stage of the studied patients.

## 6. Conclusions

In conclusion, it seems that an exercise program intervention provides benefits in psychoemotional state and QoL in adult patients with cancer and HSCT. Thus, exercise training programs may have a positive emotional effect, as well as a preventive effect on possible complications of transplantation. However, more RCTs are needed to confirm these conclusions, as some studies do not seem to show significant improvement in the variables studied due to: (a) measurement of fatigue with Multidimensional Fatigue Inventory and QLQ-C30, the latter being a less specific tool in its measurement [43]; (b) measurement of anxiety with the Profile of Mood State and Hospital Anxiety and Depression Scale with an increase in anxiety at the end of exercise in the intervention group [40]; (c) use of standardized questionnaires, such as QLQ-C30 and Functional Assessment of Cancer Therapy-Bone Marrow Transplant, instead of questionnaires focused on the individual factors of each subject [65]; and (d) psychological tests and tests that were not very new and up to date were included [83].

According to the studies analyzed, the most appropriate exercise program to improve psychoemotional variables had the following characteristics: it was supervised, with a duration of 12 weeks and sessions between 20 and 40 min. It combined strength and aerobic training and was performed in health facilities. Unfortunately, almost all the studies did not adequately report the intensity of training, so the most appropriate intensity cannot be proposed.

Our results show that exercise program interventions are clinically relevant and need to be widely implemented in these patients. Most of the programs analyzed were supervised, multicomponent (i.e., strength training and aerobic training as the primary element of training, with stretching, relaxation, and activities of daily living as secondary work), with a frequency range between 2 and 7 days a week, lasting 5 to 18 weeks, with sessions of 15 to 60 min, and performed in healthcare facilities or at the patient’s home (depending on the patient’s condition).

In addition, to maximize the emotional and QoL benefits of the program, exercise load and intensity should be individualized and monitored during the training process.

## Figures and Tables

**Figure 1 ijerph-19-15896-f001:**
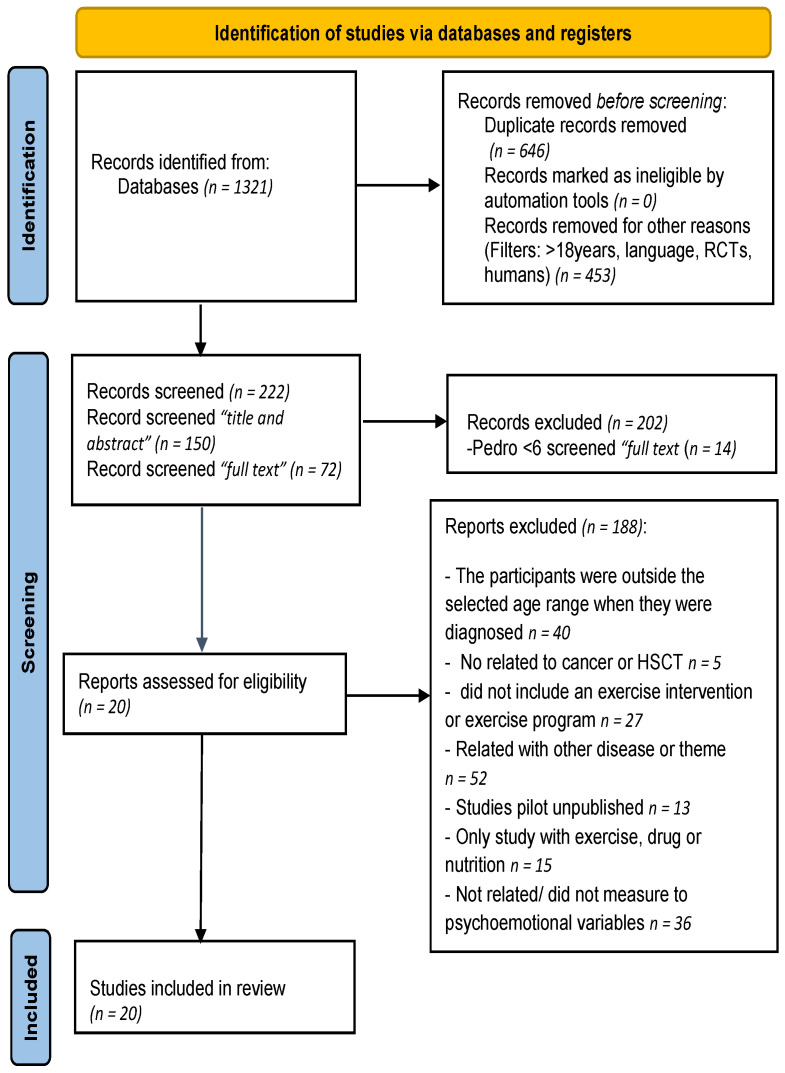
Systematic review flowchart.

**Table 1 ijerph-19-15896-t001:** Quality of the studies included in the systematic review.

Items												
Study	1	2	3	4	5	6	7	8	9	10	11	Total Score
(Knols et al., 2011) [53]	+	+	+	+	+	−	+	+	+	+	+	10
(Persoon et al., 2017) [54]	+	+	+	+	+	+	+	+	+	+	+	10
(Barğı et al., 2016) [55]	+	+	+	+	+	+	−	+	+	+	+	9
(Hacker et al., 2022) [56]	+	+	+	+	+	−	−	+	+	+	+	8
(Pahl et al., 2020) [57]	+	+	+	+	+	−	−	+	+	+	+	8
(Jarden et al., 2009) [42]	+	+	+	+	+	−	−	+	+	+	+	8
(Schumacher et al., 2018) [58]	+	+	+	+	+	?	?	+	+	+	+	8
(Wiskemann et al., 2014) [59]	+	+	+	+	+	−	−	+	+	+	+	8
(Bird et al., 2010) [60]	+	+	+	+	−	−	−	+	+	+	+	7
(Van Dongen et al., 2019) [61]	+	+	+	+	−	−	−	+	+	+	+	7
(Baumann et al., 2010) [34]	+	+	+	+	−	−	−	+	+	+	+	7
(Wiskemann et al., 2011) [40]	+	+	+	+	?	?	?	+	+	+	+	7
(Hacker et al., 2011) [41]	+	+	+	+	?	?	?	+	+	+	+	7
(Shelton et al., 2009) [62]	+	+	+	+	−	−	−	+	+	+	+	7
(Baumann et al., 2011) [63]	+	+	+	+	−	−	−	+	+	+	+	7
(Coleman et al., 2003) [64]	+	+	+	+	?	?	?	+	+	+	+	7
(Jarden et al., 2009) [65]	+	+	+	+	−	−	−	+	+	+	+	7
(Potiaumpai et al., 2021) [66]	+	+	+	+	−	−	−	+	+	+	+	7
(Oechsle et al., 2014) [43]	+	+	−	+	−	−	−	+	+	+	+	6
(Defor et al., 2007) [67]	+	+	+	?	?	?	?	+	+	+	+	6

Column numbers correspond to the following criteria on the PEDro scale: 1—Eligibility criteria were specified. 2—Subjects were randomly allocated to groups (or in a crossover study, subjects were randomly allocated an order in which treatments were received. 3—Allocation was concealed. 4—Groups were similar at baseline. 5—Subjects were blinded. 6—Therapists who administered the treatment were blinded. 7—Assessors were blinded. 8—Measures of key outcomes were obtained from more than 85% of subjects. 9—Data were analyzed by intention to treat. 10—Statistical comparisons between groups were conducted. 11—Point measures and measures of variability were provided. A total score out of 10 is determined from the number of criteria that are satisfied, except that scale item 1 is not used to generate the total score. + Indicates the criterion was clearly satisfied, − indicates that it was not, and ? indicates that it is not clear whether the criterion was satisfied.

**Table 2 ijerph-19-15896-t002:** Studies that have analyzed the effects of an exercise program before and after HSCT, with the main results in quality of life and other psychoemotional-related variables. (All the studies were ordered by PEDro score from largest to smallest).

Study	Study Design	Sample Size by Group (Sex), Age (Mean ± SD; Range), Sample Size	Primary Cancer	Intervention	Main Results in Psychoemotional Variables. Tests/Scales	Main Results in QoL Variables. Tests/Scales
(Hacker et al., 2022) [56]	RCT T1: Pre HSCT (A) T2: Post HSCT (7 weeks D) PEDro score 8	EXP: n = 17 (8 female), 62.21 ± 8.50 ^b^ years CT: n = 15 (3 female), 63.44 ± 6.30 ^b^ years	MM	Type: AT, ST Duration: 6 weeks Intensity and volume: AT 7 days a week (increase steps by 10% weekly), ST 5 Ib (female) 8 Ib (male) Supervised: Yes Setting: Home/hospital	T1-T2: • CFS: -Mental fatigue: ↓EXP/↓CT • PROMIS subscale: -Anger: ↓EXP/↓CT	T1-T2: • QLQ C-30: -EF: ↑EXP/↑CT
(Pahl et al., 2020) [57]	RCT T1: Pre HSCT (A) T2: Post HSCT (D) T3: PostHSCT (180 days D) PEDro score 8	EXP: n = 18 (7 female), 55 years (50–63) CT: n = 26 (7 female), 56 years (32–63)	ALL, AML, CLL, CML, lymphoma, MDS, MF, MM, SAA, septic granulomatosis	Type: AT, ST with vibration (EXP), mobilization of the spine and stretching (CT) Duration: N/R Intensity: N/R Volume: AT/WBV/ST (EXP) 5 days a week (20 min), mobilization and stretching (CT) 5 days a week (20 min) Supervised: Yes Setting: Hospital	T2-T3: -MFI: ↓EXP/↑CT	T1-T3: -QLQ-C30: • QOL: ↓CT • PF: ↓CT T2-T3: -QLQ-C30: • QOL: ↑EXP/↑CT
(Jarden et al., 2009) [42]	RCT T1: Pre HSCT (A) T2: Post HSCT (6 weeks) PEDro score 8	EXP: n = 21 (8 female), 40.9 years (18–60) CT: n = 21 (8 female), 37.4 years (18–55)	AA, ALL, AML, CML, MDS, MF, PNH, WM	Type: AT, ST, stretching, relaxation Duration: 4–6 weeks Intensity and volume: AT 5 days a week (HRmax 50–75% low to moderate RPE 10/13), stretching (dynamic: 1–2 sets of 10–12 reps; static: 1 set/15–30 seg), ST 3 days a week (1–2 sets of 10–12 reps at low to moderate, RPE 10/13) and relaxation twice a week (low RPE 6/9) Cadence: 30–70 cycles/min and range at 30–75 W Supervised: Yes Setting: Hospital	T1-T2: -HADS: =EXP/=CT	T1-T2: -FACT-BMT: =EXP/=CT -KPS: =EXP/=CT -QLQ-C30: =EXP/=CT
(Wiskemann et al., 2014) [59]	Multicenter RCT T1: Pre HSCT (baseline)—pre HSCT (A) T2: Pre HSCT (A)–post HSCT (D) T3: Post HSCT (D)–post HSCT (6–8 weeks) PEDro score 8	EXP: n = 52 (21 female), 47.6 years (18–70) CT: n = 53 (13 female), 50 years (20–71)	AA, ALL, AML, CLL, CML, Lymphoma, MDS, MM	Type: AT, ST Duration: 8 weeks Frequency: AT: T1 (3 days a week), T2 (5 days a week), T3 (3 days a week). ST: T1, T2, and T3 (twice a week). Intensity and volume: Not specified Supervised: Yes Setting: Home/hospital	T1-T3: -MFI: =EXP/=CT -DT: =EXP/=CT	T1-T3: -QLQ-C30: =EXP/=CT
(Baumann et al., 2010) [34]	RCT T1: Pre HSCT (A) T2: Post HSCT (D) PEDro score 7	EXP: n = 32 (11 female), 44.9 ± 12.4 ^b^ years CT: n = 32 (18 female), 44.1 ± 14.2 ^b^ years	ALL, AML, CLL, CML, LHN, MDS, MM, Solid tumor, immuno-deficiency	Type: AT, ADL Duration: 7 weeks Volume: AT twice a week (80% HRmax), ADL 5 days a week (5 × 20 steps with 1 min break RPE “slightly strenuous” to ‘strenuous’. CT 5 days a week Cadence: AT (increase 25 W/2 min) Supervised: Yes Setting: Hospital	-No measurement	T1-T2: -QLQ-C30: • PF: ↓EXP/↓CT • RF: ↓EXP/↓CT • GQOL: ↓CT • Cognitive functioning: ↓CT • Fatigue: ↑CT • Pain: ↑EXP/↑CT • Sleep disturbance: ↑CT
(Wiskemann et al., 2011) [40]	RCT T1: Pre HSCT (medical checkup)—pre HSCT (4 week A) T2: Post HSCT (H)—post HSCT (D) T3: Post HSCT (D)—post HSCT (D 6–8 weeks) PEDro score 7	EXP: n = 52 (21 female), 47.6 years (18–70) CT: n = 53 (13 female), 50 years (20–71)	AA, ALL, AML, CLL, CML, MDS, MM, MPS, Others	Type: AT, ST Duration: 18 weeks Intensity and volume: AT 1–4 week to (A) 3 days a week, from (H) 3–5 days a week RPE (12–14/20), DCT: color codes (red 15–20 min, yellow 20–30 min, green 30–40 min), from 1–8 week rehabilitation 3 days a week. ST 1–4 week to (A) twice a week, from (H) twice a week (2–3 sets of 8–20 reps RPE 14–16/20), from 1–8 week rehabilitation twice a week. Supervised: Yes T2 and self-directed T1 and T3 Setting: Home/hospital	T2: -MFI: GF (↑EXP), Pf (↑EXP) -POMS: Fatigue (↓EXP), Anger/hostility (↓EXP) T3: -MFI: GF (↓CT), Pf (↓CT), -HADS: Anxiety (↑EXP) -POMS: Depression (↓EXP), fatigue (↑CT), anger/hostility (↑CT)	T3: -QLQ-C30: PF (↓EXP), pain (↑EXP)
(Baumann et al., 2011) [63]	RCT T1: Pre HSCT(A) T2: Post HSCT(7–8 weeks) PEDro score 7	EXP: n = 17 (6 female), 41.41 ± 11.78 ^b^ years CT: n = 16 (11 female), 42.81 ± 14.04 ^b^ years	ALL, AML, CLL, CML, MDS, MM, MPS, PID	Type: AT, ADL, stretching, coordination Duration: 7–8 weeks Volume and intensity: AT (H) once–twice a week (10–20 min/day uninterrupted or interval training at HRmax 80%); ADL-training (H) 5 days a day (20 min at day, 5 × 20 steps with 1 min break of slightly strenuous or strenuous); mobilization until 1 day before discharge daily except on weekend low intensity; or not strenuous (CT) 20 min at day Cadence: AT cycle (since 25 W with 25 W increment every 2 min) Supervised: Yes Setting: Hospital	-No measurement	T1-T2: -QLQ-C30: Fatigue (↑CT); PF (↓EXP/↓CT); emotional state (↑EXP)
(Coleman et al., 2003) [64]	RCT with RM T1:Pre HSCT (A) T2: Post HSCT (3 months) PEDro score 7	EXP: n = 14 CT: n = 10 (10 female), 55 years (42–74)	MM	Type: AT, ST Duration: 6 months Intensity and volume: AT 3 days a week CT (18 min fast paced walking at RPE 12–15/20), ST 3 days a week with color bands (1 set of 8 red 9–15 Ib, 1 set of 8 green 5–9 Ib) and (2 sets of 8 chair stands of 1 RM) Supervised: Not Setting: Home	T1-T2: -POMS: ↓EXP/↓CT; fatigue: ↓EXP	
(Jarden et al., 2009) [65]	RCT T1: Pre HSCT (A) T2: PostHSCT (D) PEDro score 7	EXP: n = 21 (8 female), 45.0 years (18–60) CT: n = 21 (8 female), 38.0 years (18–55)	AA, ALL, AML, CML, MDS, MF, PNH, WM	Type: AT, ST, stretching, relaxation Duration: 4–6 weeks Intensity and volume: AT 5 days a week low to moderate (HRmax 50–75% of RPE 10/13), stretching (dynamic: 1–2 sets of 10–12 reps; static: 1 set/15–30 sg), ST 3 days a week low to moderate (1–2 sets of 10–12 reps at RPE 10/13) and relaxation twice a week (Low RPE 6/9) Cadence: 30–70 cycles/min and range of 30–75 W Volume: ST, stretching, and relaxation (dynamic: (1–2 sets of 10–12 reps); static: (1 set 15–30 sg)) Supervised: Yes Setting: Hospital	T1–T2: -No measurement.	T1–T2: - FACT-BMT: = EXP/ = CT - KPS: = EXP/ = CT - QLQ-C30: = EXP/ = CT
(Potiaumpai et al., 2021) [66]	RCT T1: Pre HSCT (1–3 days A) T2: Pre HSCT (3–5 days D) T3: Post HSCT (30 days D) PEDro score 7	EXP: n = 19 (8 female), 59.3 ± 7.9 ^b^ years CT: n = 16 (11 female), 58.2 ± 7.4 ^b^ years	ALL, AML, CLL, MDS, MM, Others	Type: AT Duration: 30 days Intensity and volume: AT 3 days a week (5–30 min at RPE 7–8/13) Supervised: Yes Setting: Hospital	- No measurement	T2-T3 - QOLS: ↑ EXP - FACT-BMT: ↑ EXP
(DeFor et al., 2007) [67]	RCT T1: Pre HSCT (A) T2: Post HSCT (100 days) PEDro score 6	EXP: n = 51 (22 female), 46 years (18–68) CT: n = 49 (17 female), 49 years (22–64)	AA, ALL, AML, CML, HL, LHN, MDS	Type: AT Duration: 100 days Intensity: Comfortable speed Supervised: Not Frequency: 7 days/week Setting: Clinic/home	T1–T2: - Emotional Score: ↑ EXP/↓ CT	T1-T2: - KPS: ↑ EXP

Abbreviations: A = Admission; AA = Aplastic anemia; ADL = Activities of daily living; ALL = Acute lymphoid leukemia; AML = Acute myeloid leukemia; AT = Aerobic training; CLL = Chronic lymphocytic leukemia; CML = Chronic myeloid leukemia; CT = Control group; D = Discharge; DCT = daily cardiovascular training; EXP: Experimental group; EORCT-QOL = Questionnaire developed to assess the quality of life of cancer patients; FACT- BMT = Functional Assessment of Cancer Therapy-Bone Marrow Transplant; GF = General fatigue; GQOL = Global quality of life; H = Hospitalization; HADS = Hospital Anxiety and Depression Questionnaire; HL = Hodgkin’s lymphoma; Ib = Pounds; KPS = Karfnosky performance scale; LHN = Non-Hodgkin’s lymphoma; HSCT = Hematopoietic stem cell transplantation; HRmax = Heart rate maximal; MDS = Myelodysplastic síndrome; MF = Myelofibrosis; MFI = multidimensional fatigue inventory; MM = Múltiple mieloma; MPS = Myeloproliferative síndrome; PF = Physical functioning; Pf = Physical fatigue; PNH = paroxysmal nocturnal hemoglobinuria; POMS = Profile of Mood States; PID = primary immune deficiency; QOL = Quality of life; RCT = randomized controlled trial; RF = Role functioning; RM = repeated measures; RPE = Rate of Perceived Exertion; SAA = Severe aplastic anemia; ST = Strength training; WBV = Whole body vibration; WM = Waldenstrom macroglobulinemia. Symbols: ↑ = increase; ↓ = decrease; n = Sample size; N/R = not reported; Reps= repetition; W = Watios; ^b^ (Mean ± SD).

**Table 3 ijerph-19-15896-t003:** Studies that have analyzed the effects of an exercise program after HSCT, with the main results in quality of life and other psycho-emotional related variables. (All the studies were order by PEDro score from largest to smallest).

Study	Study Design	Sample Size by Group (Sex), Age (Mean ± SD; Range)	Primary Cancer	Intervention	Main Results in Psychoemotional Variables	Main Results in QoL Variables
(Knols et al., 2011) [53]	RCT T1: Pre HSCT (A)—post HSCT (D) T2: Post HSCT (D)—post HSCT (3 months) PEDro score 10	EXP: n = 64 (26 female), 46.7 ± 13.7 years (18–75) ^a^ CT: n = 67 (28 female), 46.6 ± 12 years (20–67) ^a^ Sample Size: d = 0.5	ALL, AML, amyloidosis, CLL, HL, LHN, lymphoma, MM, osteomyelofibrosis, testicular cancer	Type: AT, ST Frequency: 2 days/week Duration: 12 weeks Intensity and volume: AT twice a time (50–70% to 80% HRmax) Supervised: Yes Setting: Fitness center/physiotherapy practice	-No measurement	T1-T2: -HRQOL: Emotional function: ↑EXP/↑CT
(Persoon et al., 2017) [54]	RCT T1: Post HSCT (A) T2: Post HSCT (18 weeks) PEDro score 10	EXP: n = 54 (22 female), 53.5 years (20–67) CT: n = 55 (18 female), 56 years (19–67)	HL, MM	Type: AT, ST Duration: 18 weeks Intensity and volume: AT 1–8 weeks twice a week (blocks of 30 sg at 65% MSEC alternated with blocks of 60 s at 30% MSEC), from 9 to 18 weeks (blocks of 30 s at 65% MSEC alternated with blocks of 30 sg at 30% MSEC), ST 1–12 weeks twice a week (2 sets/10 reps 60–80% 1 RM, from 13–18 weeks once a week (1 set/20 reps 35–40% 1 RM). Supervised: Yes Setting: Local physiotherapy	T1–T2: -No changes	T1–T2: -No changes
(Bargi et al., 2016) [55]	RCT T1: Pre HSCT (A) T2: Post HSCT (6 weeks) PEDro score 9	EXP: n = 20 (8 female), 34.10 ± 12.65 ^a^ years CT: n = 18, (6 female), 39.11 ± 12.57 ^a^ years	AA, ALL, AML, CML, Fanconi anemia, MDS, MM, LHN, PNH	Type: AT and respiratory muscle Duration: 6 weeks Volume and intensity: AT: 7 days a week (speed progressively increased at 1 min intervals walking at 12 stages/30 min rest between 2 test with FIS (1–4). Diaphragmatic breaths: 7 days a week (EXP) (15 sg/25–30 breaths/5–10 resting IMT at 40% of MIP), (CT) (received sham IMT at fixed workload, 5% of baseline MIP with MMRC (0–4)) Supervised: Yes Setting: Hospital/home	T1–T2: -FIS: ↓EXP (Effect size: d = −0.27) -MADRS: ↓EXP	T1–T2: -QLQ-C30: • Global health status: ↑EXP (effect size: d = 0.39) • Functional scale scores: ↑EXP (effect size: d = 0.69)
(Schumacher et al., 2018) [58]	RCT T1: Pre HSCT (A)—post HSCT (14 days) T2: Pre HSCT (A)—post HSCT (100 days) PEDro score 8	EXP: n = 19 (3 female), 56 years (21–65) ^b^ CT: n = 23 (14 female), 56.5 years (23–69) ^b^	AML, CLL, CML, LHN, MDS, MM, teratoma	Type: AT, ST, stretching, Wii sports, Wii fit program, Wii balance Duration: 100 days Intensity: N/R Frequency: 5 days/week Supervised: Yes Setting: Hospital	T1–T2: -SF36: Vitality: ↑EXP -HADS: ↓EXP/↓CT -Distress thermometer: ↑EXP	T1–T2: -FACT-BMT: • PWB: ↑EXP/↑CT • EWB: ↑EXP • FWB: ↑EXP
(Bird et al., 2010) [60]	RCT T1: Post HSCT (A) T2: Post HSCT(6 months) PEDro score 7	EXP: n = 29 (13 female), 57 years (44–53.5) CT: n = 29 (7 female), 52 years (42.5–63)	Leukemia, lymphoma, myeloma	Type: AT, relaxation Duration: 10 weeks Intensity and volume: AT (EXP) 1–10 week (a series of circuit training exercises), relaxation (guided imagery). AT (CT) 1–10 weeks 3 days a week (home-based exercise program) Supervised: Yes Setting: Hospital/home	T1–T2: -SF36: =EXP/=CT	T1–T2: -QoL: =EXP/=CT - GHQ-12: =EXP/=CT
(Van Dongen et al., 2019) [61]	Multicenter RCT T1: Post HSCT (baseline) T2: Post HSCT (after exercise or similar time point in the CT) T3: Post HSCT (12 months later) PEDro score 7	EXP: n = 54 (22 female), 52 ± 11 ^b^ years CT: n = 55 (2 female), 53 ± 12 ^b^ years	HL, MM	Type: AT, ST Duration: 18 weeks Intensity and volume: AT 1–8 weeks twice a week (2 × 8 min, alternating 30 sg at 65% and 60 sg at 30% MSEC), 9–12 weeks twice a week (2 × 8 min, alternating 30 sg at 65% and 30 sg at 30% MSEC), 13–18 weeks once a week (2 × 8 min, alternating 30 s at 65% and 30 sg at 30% MSEC), ST: 1–12 weeks twice a week (2 sets of 10 reps at 65–80% of 1-RM), 13–18 week once a week (2 sets of 20 reps at 35–40% of 1-RM) Supervised: Yes Setting: Hospital	T1–T3: -MFI: =EXP/=CT	T1–T3: -HRQoL: =EXP/=CT
(Hacker et al., 2011) [41]	RCT T1: Pre HSCT—post-HSCT (after 8 days) T2: H-post HSCT (1–6 weeks) PEDro score 7	EXP: n = 9 CT: n = 10 (5 female), 46.26 years (16.23) ^b^	AML, lymphoma	Type: ST Duration: 6 weeks Volume and intensity: ST 1–6 week 3 days a week (1–2 sets of 8–10 reps of RPE (13/20) Supervised: Yes Setting: Hospital/home	T2: -No measurement	T2: -QLQ-C30: =EXP/=CT -QLI: =EXP/=CT
(Shelton et al., 2009) [62]	RCT T1: Pre HSCT (A) T2: Post HSCT (4 weeks) PEDro score 7	Supervised: n = 26 (9 female), 43.65 ±13.18 ^a^ years Self-directed: n = 27 (11 female), 48,93 ± 11.66 ^a^ years	ALL, AML, CLL, CML, HD, LHN	Type: AT, ST Duration: 4 weeks Volume and intensity: AT: 3 days a week (20– 30 min 60–75% HR max and BFI: 0–10), ST: 3 days a week EXP supervised (1–3 sets of 10 reps), EXP self-directed (1–3 sets of 10–15 reps). The AT and ST increased every third visit, if extreme fatigue, resistance was reduced to the previous level. Supervised: Yes Setting: Hospital/home	T1–T2: -BFI: =EXP/=CT	-No measurement
(Oechsle et al., 2014) [43]	RCT T1: Post HSCT (A) T2: Post HSCT (after intervention) PEDro score 6	EXP: n = 17 (7 female), years 51.7 ± 13.3 ^b^ CT: n = 17 (7 female), years 52.9 ± 15.4 ^b^	AML, LHN, MM, germ cell tumor	Type: AT, ST Duration: 21 days Intensity and volume: AT 5 days a week (10–20 min), ST 5 days a week (20 min, 2 sets of 16–25 reps at 40–60% of 1 RM) Rest: AT (regular pauses until recuperated to 66.6%) Supervised: Yes Setting: Hospital	T1–T2: - MFI: • Cognition: ↑CT Psychosocial function: ↑CT	T1–T2: -QLQ-C30: • PF: ↑EXP

Abbreviations: A = admission; AA = aplastic anemia; ALL = acute lymphoid leukemia; AML = acute myeloid leukemia; AT = aerobic training; BFI = brief fatigue inventory; CLL = chronic lymphocytic leukemia; CML = chronic myeloid leukemia; CT = control group; D = discharge; EXP = experimental group; QLQ C-30 = questionnaire developed to assess the quality of life of cancer patients; EWB = emotional well-being; FACT-BMT = Functional Assessment of Cancer Therapy-Bone Marrow Transplant; FIS = fatigue impact scale; FWB = functional well-being; GHQ-12 = General Health Questionnaire; HD = Hodgkin’s disease; HADS = Hospital Anxiety and Depression Questionnaire; HL = Hodgkin’s lymphoma; IMT = inspiratory muscle training; QLI = quality of life index; LHN = non-Hodgkin’s lymphoma; HSCT = hematopoietic stem cell transplantation; HRmax = heart rate maximal; HRQOL = health-related quality of life; MADRS = Montgomery–Âsberg Depression Rating Scale; MDS = myelodysplastic syndrome; MFI = Multidimensional Fatigue Inventory; MIP = maximal inspiratory pressure; MM = multiple myeloma; MMRC = Modified Medical Research Council; MSEC = maximal short exercise capacity; PF = physical functioning; PNH = paroxysmal nocturnal hemoglobinuria; PWB = physical well-being; RCT = randomized controlled trial; RM = repeated measures; RPE = rate of perceived exertion; SF36 = Short Form Health Survey; ST = strength training. Symbols: ↑ = increase; ↓ = decrease; (=) = no changes; Min = minutes; n = sample size; N/R = not reported; Reps = repetition; Sg = seconds; ^a^ (median ± SD); ^b^ (mean ± SD).

## Data Availability

Not applicable.

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
