# Peer review of "Effects of Exercise Programs on Psychoemotional and Quality-of-Life Factors in Adult Patients with Cancer and Hematopoietic Stem Cell Transplantation or Bone Marrow Transplantation: A Systematic Review"

_ijerph, 2022, doi:10.3390/ijerph192315896_

Round 1

Author Response

IJERPH- 1991674 Cover letter

Effects of Exercise Programs on Psycho-emotional and Quality of life Factors in Adult Patients with Cancer, Bone marrow transplant and Hematopoietic Stem Cell Transplant: A Systematic Review.

Authors: Erica Morales Rodríguez, Jorge Lorenzo Calvo, Miriam Granado Peinado, Txomin Pérez-Bilbao and Alejandro F. San Juan.

About this cover letter

In this cover letter, we will answer the reviewers suggestions in order. On page 2, annotations from REVIEWER 1 will be answered in blue. On page 5, the REVIEWER 2 annotations will be answered in blue. On page 6, the REVIEWER 3 annotations will be answered in blue. We attach a manuscript with all the corrections. The line and table numbers have changed, so you must follow the corrected manuscript.

REVIEWER 1 ANNOTATIONS.

“L59-63: Some arguments are given in a simplistic way: cancer related fatigue should not only be opposed to normal physical fatigue but might be related to other fatigue forms like chronic fatigue syndrome”. 

Thank you very much for the suggestion. We have added a new quote to make a stronger case for the differences between cancer-related fatigue and other types of fatigue, including chronic fatigue syndrome.

“L77-78: Repeated "second-hand" citations like [28]”.

Thank you very much for your comment, this citation has been checked again with the Mendeley bibliographic manager and we have corrected the citation.

“L 79-87: Only a very short passage summarizes the effects of exercise programs in the patient group of interest. A short view on other patient groups might be worthwhile”.

Thank you very much for your comment, we have elaborated briefly on this section. However, this information is more fully developed in the study "Effects of Exercise Programs on Physical Factors and Safety in Adult Patients with Cancer and Haematopoietic Stem Cell Transplantation: A Systematic Review".

“L313-316: However, the advantage of including only methodological sound publications results in a very high selection (653 studies from 673 studies were excluded). Should be discussed with respect to”.

Thank you very much for your comment. We have added new information in the section 4. Discussion”.

“L180-Table 2 is hardly readable due to a large list of abbreviations. The chronological order might be questioned”.

“Characteristics of interventions might be switched to the text and be discussed in more detail with respect to the outcomes of the studies. No details of the different studies discussed, no hints for positive, negative and no effects. This should be added, as it is not a meta analysis”. 

Thank you for your suggestions. The characteristics of the interventions are developed in section 3.4. Characteristics of exercise interventions. Likewise, in section 3.6. Endpoints and Exercise Intervention Psycho-emotional results indicates the studies in which significant improvements are produced. Finally, in section 4. Discussion, the positive effects found are discussed, adding the studies that do not find improvement and trying to explain this occurrence.

“L167-179: Control conditions might need more attention”.

Thanks for your comment. We have added a paragraph on control conditions.

“Positive and negative effects are not clearly separated as an increase in fatigue is not a positive intervention effect”.

Thank you for your comment. We have revised section 3.6. Assessment criteria and -psycho-emotional outcomes of the exercise intervention, where the negative and positive aspects have been differentiated by sections.

“The authors might wish to have a look at the variability of effects and the effect sizes. Reliability of variables not considered”.

Thank you for your suggestion. We have included "effect sizes" in the results tables of those studies that clearly report them.

“L167-179 Changes in Control groups not discussed (negative changes, activities in control groups)”.

Thanks for your comment. We have amplified the conditions of the control group. We have better described what care they receive and whether they have been given general exercise recommendations.

“No performance aspects considered or discussed, which might be probable as fatigue increases”.

Thank you for your comment. We have not discussed the performance issues about the fatigue in the systematic review because the purpose of the review the purpose of this manuscript is to systematically review the current literature and analyses the scientific evidence that have examined the effects of exercise programs on psycho-emotional and QoL factors in adult patients with cancer and HSCT.

“L395: The authors claim that randomized baseline fitness should be considered. As this is directly dependent on severity of disease symptoms this might be the better argument”.

Thank you very much for your comment. We have completed the sentence with: No randomization of baseline fitness characteristics prior to exercise intervention, independently of severity of pathology or symptomatology. 

“L334: Exclusion of studies with nutrition intervention is critical point, as this is not confounding variable but an intervening variable”.

I gratefully your comment. This is an intervention variable that we decided not to include as it would further limit the number of studies found and we wanted to focus exclusively on psycho-emotional and quality of life intervention variables. In addition, no author related to this knowledge has participated in the study. However, we have changed the term "confounding variable" to "intervening variable".

“L425: Claiming the need of larger samples sizes might be questioned, as high quality experimentally oriented studies will provide large sample sizes only in very large multi-center studies. Lot of worthless internet surveys with large sample sizes might serve as warning sign for this argument”.

Thank you very much for your comment. We have reviewed the future lines of research and have removed this phrase: "larger sample size".

“L407-411 Finally, the authors might like to give at least a brief argument why performance and performance deficits (physical and mental) are excluded as this is very important for quality of life”.

Your comment is greatly appreciated.  We have introduced in the final section of the discussion several systematic reviews on improvements in health and fitness. We have also added a systematic review about mental performance.

Reviewer 2 Report

Dear authors.

I would like to congratulate you for the development of this manuscript entitled “Effects of Exercise Programs on Psycho-emotional and Quality of life Factors in Adult Patients with Cancer and HSCT: A Systematic Review" that tries to analyse the effects of the application of exercises on psycho-emotional and quality of life (QoL) factors  in adult patients with cancer and hematopoietic stem cell transplant (HSCT) or bone marrow transplant (BMT). For this, 20 RCTs of proven quality were selected, obtaining a total sample of 1219 participants.

The study has an introduction that clearly justifies the need for the study, as well as clearly presenting its objective.

The methodology follows an order that facilitates reading.

Regarding the results, the tables are well organized and present the information that is considered relevant for the study.

The discussion is well structured, grounded and oriented to the objectives previously established in the study.

Limitations are well categorized and presented in an orderly manner.

The conclusions are obtained based on what was previously described in the manuscript.

The entire structure of the manuscript is considered correct and also interesting. I would simply like to ask the authors, which professionals should be in charge of carrying out the implementation and supervision of these exercise programs, especially because of what is stated in the last paragraph of the conclusion.

On the other hand, the manuscript must be revised because there are blank spaces that sometimes make it difficult to read.

Thank you very much and congratulations for this work.

Author Response

IJERPH- 1991674 Cover letter

Effects of Exercise Programs on Psycho-emotional and Quality of life Factors in Adult Patients with Cancer, Bone marrow transplant and Hematopoietic Stem Cell Transplant: A Systematic Review.

Authors: Erica Morales Rodríguez, Jorge Lorenzo Calvo, Miriam Granado Peinado, Txomin Pérez-Bilbao and Alejandro F. San Juan.

About this cover letter

In this cover letter, we will answer the reviewers suggestions in order. On page 2, annotations from REVIEWER 1 will be answered in blue. On page 5, the REVIEWER 2 annotations will be answered in blue. On page 6, the REVIEWER 3 annotations will be answered in blue. We attach a manuscript with all the corrections. The line and table numbers have changed, so you must follow the corrected manuscript.

“I would simply like to ask the authors, which professionals should be in charge of carrying out the implementation and supervision of these exercise programs, especially because of what is stated in the last paragraph of the conclusion”.

In our opinion, it should be a multidisciplinary team in charge of implementing and supervising the exercise programmes. This team should be composed of: graduates in physical activity and sport, clinical psychologists and sports nurses.

“On the other hand, the manuscript must be revised because there are blank spaces that sometimes make it difficult to read”.

We have revised the manuscript again and removed the blank spaces to make it easier to read.

Reviewer 3 Report

Aim of study: “…the purpose of this manuscript is to systematically review the current literature and analyze the scientific evidence that have examined the effects of exercise programs on psycho-emotional and QoL factors in adult patients with cancer and HSCT.”

My major concern for the current manuscript is that there are 3 similar systematic reviews and meta-analyses published recently (2021 and 2022 - see references 60-62), and it is unclear what the current review adds to those findings. Further the previous three studies included a meta-analysis approach which provides an additional quantitive component that the current review does not.

Paper should be revised by a native English speaker.

Title: avoid use of abbreviations.

Abstract 

- “adult patients with cancer and hematopoietic stem cell transplantation (HSCT) or bone marrow transplantation (BMT).”  while the title says with cancer and HSCT only. The authors should be be consistent.

- Line 22 - randomized studies or randomized controlled studies. Please review.

- Line 25-26 - QoL has already been defined, yet the authors write quality of life. Please review the use of abbreviations throughout the manuscript. 

- Line 27 - missing punctuation.

Introduction

- Line 64 - QoL has already been defined, yet the authors write quality of life. Please view use of abbreviations throughout the manuscript. 

Methods

- Line 99 should be “The systematic review is reported in accordance with the Preferred Reporting “ 

- Line 115 - PEDRO has not been defined in the manuscript text. Why did the authors exclude studies with PEDRO<7? I believe the authors could have included all studies regardless of the PEDRO score and discussed the potential impact of including studies with poor/fair quality.

- The authors should conduct at least backward citation of included studies to ensure additional potential studies that could have been missed through database searches are reviewed, and they should also review the reference lists of relevant systematic reviews and meta-analysis.

- The authors do not adequately report step 1 (screening of title and abstract) and step 2 (screening of full text) revision of studies and exclusions.

- Line 115 and lines 126-132 - How did the authors exclude studies with PEDRO<7? When were studies scored according to the PEDRO scale? How many studies were excluded due to this?

- Table 1 should be presented in the results. 

- Outcomes subsection in the methods - the authors are describing results from the papers included in the systematic review. Most of this information belongs in the results section of the manuscript. Please revise accordingly.

- Why did the authors opt to not conduct a meta-analysis? Previous studies (as listed in line 307) included systematic review and meta-analysis.

Results

- Figure 1 and Lines 170-172 - It appears that the authors excluded 653 studies during the screening of the title and abstract, however they state 20 reports were assessed for eligibility and provide reasons for exclusion for 653 studies and then arrive at 20 studies included in the review. This does not make sense. The authors must specify how many studies were excluded following title and abstract screening, and then how many studies were excluded following full text review (with reasons).

The inclusion criteria described in lines 110-118 do not match those provided as exclusion criteria in Figure 1. The criteria should be consistent.

- Lines 218-227 - it is sometimes unclear whether the studies did not provide information for the type of intensity training or if they did not include specific types of intensity training as part of their exercise interventions. Please review.

- Line 220 - Please review - the last part of the sentence seems incomplete.

- Line 232 - Pleas review - sentence seems incomplete. 

- Lines 271-279 and 292-300 - The aim of the study is to examined the effects of exercise programs on psycho-emotional and QoL factors in adults with cancer. However, in reading the paragraph, it appears the authors provide the scales used to evaluate psycho-emotional (repetition of information presented in lines 261-270) and QoL factors (repetition of information presented in lines 282-191) instead of describing in detail the associations or effects of exercise program on these factors. I wonder if there are specific factors that improve or deteriorate with exercise programs during cancer treatment. For example, it is unclear how anxiety and depression are impacted. Further, are there specific constructs within the 26-item short form survey, QoL (functional, symptoms, or global scores) that improved? I believe the authors could revise these two paragraphs of the results. Further were there any particular exercise programs (intensity, duration etc) that significantly improved psycho-emotional and QoL factors?

- I believe the authors should add information on when psycho-emotional and QoL factors were evaluated to the table as well as the results.

Discussion

- It is unclear from the results how the authors arrived at the main findings reported in lines 303-306. The results must provide information that support the main findings.

- Lines 324-328 and 363-367 - I believe this information should be provided in the results.

- A few of the paragraphs in the discussion provide lists organized by letters. I believe the authors should try to avoid this. Please consider revising the discussion accordingly.

- The authors provide information on three systematic reviews and meta-analyses conducted recently. I believe the findings of these studies should be provided in the introduction of the current study. Further, the authors must clearly show what the current study adds to the existing literature.

Author Response

IJERPH- 1991674 Cover letter

Effects of Exercise Programs on Psycho-emotional and Quality of life Factors in Adult Patients with Cancer, Bone marrow transplant and Hematopoietic Stem Cell Transplant: A Systematic Review.

Authors: Erica Morales Rodríguez, Jorge Lorenzo Calvo, Miriam Granado Peinado, Txomin Pérez-Bilbao and Alejandro F. San Juan.

About this cover letter

In this cover letter, we will answer the reviewers suggestions in order. On page 2, annotations from REVIEWER 1 will be answered in blue. On page 5, the REVIEWER 2 annotations will be answered in blue. On page 6, the REVIEWER 3 annotations will be answered in blue. We attach a manuscript with all the corrections. The line and table numbers have changed, so you must follow the corrected manuscript.

REVIEWER 3 ANNOTATIONS.

“L321-337 My major concern for the current manuscript is that there are 3 similar systematic reviews and meta-analyses published recently (2021 and 2022 - see references 60-62), and it is unclear what the current review adds to those findings. Further the previous three studies included a meta-analysis approach which provides an additional quantitive component that the current review does not.”

Thank you very much for your valuable suggestion. We have added a new sentence for clarification purposes. Also, we have added the date of revision of the bibliography of each article, differentiating it from the final date of publication. We have also reduced the number of systematic reviews to three. With respect to the following studies:

  • The effect of exercise and nutrition interventions on physical functioning in patients undergoing haematopoietic stem cell transplantation: a systematic review and meta-analysis. Support Care Cancer [Internet]. 2021 [cited 2022 Feb 16];29(11):7111–26. Available from: https://doi.org/10.1007/s00520-021-06334-2.

The aim of this systematic review and meta-analysis is to determine the effect of exercise and nutrition interventions to improve physical functioning in patients receiving HSCT. They include assessment of quality of life and fatigue, but do not consider psycho-emotional variables. It is a review from 1st January 2010 to 1st July 2020.

  • Abo S, Denehy L, Ritchie D, Lin K-Y, Edbrooke L, McDonald C, Granger CL. People With Hematological Malignancies Treated With Bone Marrow Transplantation Have Improved Function, Quality of Life, and Fatigue Following Exercise Intervention: A Systematic Review and Meta-Analysis. Phys Ther [Internet]. 2021 Aug 1 [cited 2022 Feb 16];101(8). Available from: https://academic.oup.com/ptj/article/doi/10.1093/ptj/pzab130/6275370.

This systematic review aimed to assess the effect of exercise training on quality of life and fatigue outcomes. It discusses anxiety and depression in the results, but does not collect more psycho-emotional variables. Moreover, our review collects RCTs exclusively, while this review does not. It is a review from inception to 5 December 2020.

  • Xu W, Yang L, Wang Y, Wu X, Wu Y, Hu R. Effects of exercise interventions for physical fitness, fatigue, and quality of life in adult hematologic malignancy patients without receiving hematopoietic stem cell transplantation: a systematic review and meta-analysis [Internet]. Vol. 30, Supportive Care in Cancer. 2022 [cited 2022 Aug 23]. p. 7099–118. Available from: https://doi.org/10.1007/s00520-022-07029-y

            The aim of this study was to examine the effects of exercise interventions on  

            fatigue and quality of life in adults with haematological malignancies who

            did not receive haematopoietic stem cell transplantation. Our study, on the

            other hand, analyses the same effects reflecting patients before and after

            transplantation and on the other hand patients after transplantation. It is a  

            review from inception to March 2021. Even so, we have removed this study.

“Paper should be revised by a native English speaker”.

Thank you very much for the recommendation. Once the study has been corrected and accepted, we will send it for review by a native English speaker.

L3 Title: avoid use of abbreviations”.

Thank you very much for your suggestion, we have removed the abbreviations from the title.

“L4 adult patients with cancer and hematopoietic stem cell transplantation (HSCT) or bone marrow transplantation (BMT).”  while the title says with cancer and HSCT only. The authors should be be consistent.”

Thank you very much for your comment. We have added the term BMT to the title.

“L 23 randomized studies or randomized controlled studies. Please review”.

Thank you for your comment. We have detected and changed the phrase to randomized controlled studies.

“L 27 QoL has already been defined, yet the authors write quality of life. Please  

  review the use of abbreviations throughout the manuscript.” 

Thank you very much for your comment. We have added the acronym.

“L 28 - missing punctuation.”

Thank you very much for your comment. We have added the sign of punctuation.

“L67 QoL has already been defined, yet the authors write quality of life. Please view use of abbreviations throughout the manuscript”. 

Thank you very much for your comment. We have added the acronym.

“L110 should be: The systematic review is reported in accordance with the Preferred Reporting. “ 

Thank you very much for your comment. We have added the correct sentence.

“L131 - PEDRO has not been defined in the manuscript text. Why did the authors exclude studies with PEDRO<7? I believe the authors could have included all studies regardless of the PEDRO score and discussed the potential impact of including studies with poor/fair quality”.

Thank you for your comment, but PEDRO has been defined in the abstract section L22. We have decided to include only studies with PEDRO>6 to increase the methodological quality of this study and make it more interesting ( L131).

L119-121 “The authors should conduct at least backward citation of included studies to ensure additional potential studies that could have been missed through database searches are reviewed, and they should also review the reference lists of relevant systematic reviews and meta-analysis”.

Thank you for your suggestion. We have realized an amplified searching, and we have included this new information in the methods of the manuscript.

L124: “The authors do not adequately report step 1 (screening of title and abstract) and step 2 (screening of full text) revision of studies and exclusions”.

Your comment is greatly appreciated.  We have added new sentence with step 1 and 2. The exclusions are reflected in the figure 1.

“L 150 - How did the authors exclude studies with PEDRO<7? When were studies scored according to the PEDRO scale? How many studies were excluded due to this?.”

Thank you very much for your comment. Only  studies with a PEDRO score <6 were excluded. As we explain before, we have decided to include only studies with PEDRO score equal or higher than 6 points to ensure an adequate methodological quality, and to reduce the risk of bias. Finally we had selected 20 studies with these inclusions criteria. If the number had been lower (e.g., <10 studies), we would have considered lowering this quality requirement. Moreover, this method was follow by numerous systematic reviews, and is a cut-point of high methodological quality, for example:

- Morales-Rodriguez E, Pérez-Bilbao T, San Juan AF, Calvo JL. Effects of Exercise Programs on Physical Factors and Safety in Adult Patients with Cancer and Haematopoietic Stem Cell Transplantation: A Systematic Review. Int J Environ Res Public Health. 2022 Jan 24;19(3):1288. doi: 10.3390/ijerph19031288. PMID: 35162312; PMCID: PMC8834842.

- Morales JS, Valenzuela PL, Herrera-Olivares AM, Baño-Rodrigo A, Castillo-García A, Rincón-Castanedo C, Martín-Ruiz A, San-Juan AF, Fiuza-Luces C, Lucia A. Exercise Interventions and Cardiovascular Health in Childhood Cancer: A Meta-analysis. Int J Sports Med. 2020 Mar;41(3):141-153. doi: 10.1055/a-1073-8104. Epub 2020 Jan 14. PMID: 31935777.

L197-“Table 1 should be presented in the results”.

Thanks for your comment. We have added a sentence in section 3.3. Quality assessment and publication bias, explaining that the full results can be found in table 1.

L260-“Outcomes subsection in the methods - the authors are describing results from the papers included in the systematic review. Most of this information belongs in the results section of the manuscript. Please revise accordingly”.

Thanks for your comment. We have added to sections 3.6.1. Psycho-emotional variables and 3.6.2. Quality of life variables the information contained in the sections 2.5.1. Psycho-emotional variables and 2.5.2. Quality of life variables.

“Why did the authors opt to not conduct a meta-analysis? Previous studies (as listed in L 317) included systematic review and meta-analysis”.

Your comment is greatly appreciated.  We haven’t conducted a meta-analysis because there are only two variables with enough studies in which the same variable has been measured using the same tool (i.e., Fatigue, and QoL). Then, we have decided to realize a systematic review(L102-L321)

“Figure 1 and L 153 – It appears that the authors excluded 653 studies during the screening of the title and abstract, however they state 20 reports were assessed for eligibility and provide reasons for exclusion for 653 studies and then arrive at 20 studies included in the review. This does not make sense. The authors must specify how many studies were excluded following title and abstract screening, and then how many studies were excluded following full text review (with reasons)”.

Thank you for your comment. We have consulted the studies in full text and detailed the flow chart further(L153).

“The inclusion criteria described in L 133-135 do not match those provided as exclusion criteria in Figure 1. The criteria should be consistent”.

Thank you very much for your valuable suggestion. We have revised "Inclusion and exclusion criteria" and adjusted the text in the exclusion criteria described in L 133-135. The exclusion criteria in Figure 1 are more specific and more developed.

“L 218 - it is sometimes unclear whether the studies did not provide information for the type of intensity training or if they did not include specific types of intensity training as part of their exercise interventions. Please review”.

Thank you very much for your comment. We have clarified the sentences written above in relation to your comments.

“L 219-221 Please review - the last part of the sentence seems incomplete”.

Thank you very much for your comment. We have revised and completed the sentence.

“L 233-234 - Please review - sentence seems incomplete”. 

Thank you very much for your comment. We have revised the sentence.

“L 271-279 and 292-300 - The aim of the study is to examined the effects of exercise programs on psycho-emotional and QoL factors in adults with cancer. However, in reading the paragraph, it appears the authors provide the scales used to evaluate psycho-emotional (repetition of information presented in lines 261-270) and QoL factors (repetition of information presented in lines 282-291) instead of describing in detail the associations or effects of exercise program on these factors. I wonder if there are specific factors that improve or deteriorate with exercise programs during cancer treatment. For example, it is unclear how anxiety and depression are impacted. Further, are there specific constructs within the 26-item short form survey, QoL (functional, symptoms, or global scores) that improved? I believe the authors could revise these two paragraphs of the results. Further were there any particular exercise programs (intensity, duration etc) that significantly improved psycho-emotional and QoL factors?”.

Thanks for your comment, but the information you are asking for is reflected in the following sections:

L372-280: They reflect the following information:

- Of the total number of studies that measured psycho-emotional and/or quality of life variables, at which time of transplantation did they show significant differences.

L262-271: They reflect the following information:

- How many studies measured the psycho-emotional variables.

- Which psycho-emotional variables were measured?

- What scales were used to measure these psycho-emotional variables?

L287-311 They reflect the following information:

- How many studies measured the QoL variables.

- What scales were used to measure these QoL variables?

“I believe the authors should add information on when psycho-emotional and QoL factors were evaluated to the table as well as the results”.

Thank you very much for your comment. The information can be found in the following tables: Table 2. Studies that have analysed the effects of exercise program before and after HSCT, with main results in quality of life and other psycho-emotional related variables. Table 3.  Studies that have analysed the effects of exercise program after HSCT, with main results in quality of life and other psycho-emotional related variables.

“It is unclear from the results how the authors arrived at the main findings reported in L 303-306. The results must provide information that support the main findings”.

Thank you very much for your comment. We have revised the content. In accordance with section "3.6.1. Psycho-emotional variables" in 57% there were significant improvements in psycho-emotional variables, being 43% exclusively after transplantation. According to section "3.6.2. Quality of life variables" there were significant improvements in the QoL variables in 53%, 47% exclusively after transplantation.

“L281 and L307 - I believe this information should be provided in the results”.

Thank you very much for your comment. This information has been moved to sections 3.6.1 Psycho-emotional variables and 3.6.2 Quality of life variables.

“A few of the paragraphs in the discussion provide lists organized by letters. I believe the authors should try to avoid this. Please consider revising the discussion accordingly”.

Thank you very much for your comment. We have organised the lists by number.

“The authors provide information on three systematic reviews and meta-analyses conducted recently. I believe the findings of these studies should be provided in the introduction of the current study. Further, the authors must clearly show what the current study adds to the existing literature”.

We have added a new sentence to the introduction section with the 2 recent systematic reviews (L102-104).

Round 2

Reviewer 3 Report

I have doubts regarding the quality of the systematic review performed by the authors considering the manner in which they report the steps, exclusions etc.

I believe the authors have not clearly described when studies were scored according to the PEDRO scale. Was this done in Step 1 (abstract screening) or Step 2 (full text screening)?

Please remove "Studies were 179 excluded until 24 August, 22. " from lines 179-180.

The authors state: "Moreover, this method was follow by numerous systematic reviews, and is a cut-point of high methodological quality, for example:" and then provide two studies by the same group of authors as examples...

Figure 1 (flow chart) continues to not make sense. The authors excluded 653 studies, and 35 full texts were screened, however the number of reports excluded with reasons exceed the 35 full texts that were screened... Please revise.

The authors are not consistent in the formatting of Pedro, PEDRO, PEDro, etc.

The authors do not describe in detail the associations or effects of exercise program on the outcomes of interest, neither do they provide information on specific factors (anxiety, depression, functional vs symptoms of QoL) that improve or deteriorate with exercise programs during cancer treatment. The authors do not adequately describe if there are any particular exercise programs (intensity, duration etc) that significantly improved psycho-emotional and QoL factors.

“A few of the paragraphs in the discussion provide lists organized by letters. I believe the authors should try to avoid this. Please consider revising the discussion accordingly”.

The authors changed the paragraphs in the discussion proving lists organized by letters to lists organized by number. I believe the authors should avoid the use of lists in the dicussion to increase readability and flow.

The authors do not clearly show what the current study adds to the existing literature when describing the previous SR on the same/similar topics.

Author Response

                                      IJERPH- 1991674 Cover letter

Effects of Exercise Programs on Psycho-emotional and Quality of life Factors in Adult Patients with Cancer, Bone marrow transplant and Hematopoietic Stem Cell Transplant: A Systematic Review.

Authors: Erica Morales Rodríguez, Jorge Lorenzo Calvo, Miriam Granado Peinado, Txomin Pérez-Bilbao and Alejandro F. San Juan.

About this cover letter

In this cover letter, we will respond to the reviewer's suggestions in order. On page 2, REVIEWER 3's annotations will be answered in blue. We enclose a manuscript with all corrections and welcome the reviewer's comments. Line and table numbers have changed, so please follow the corrected manuscript. REVIEWER 3 ANNOTATIONS.

I have doubts regarding the quality of the systematic review performed by the authors considering the manner in which they report the steps, exclusions etc.

Thank you for your comment. We have improved, ordered and clarified the section "2.2. Selection of studies with this information":

Studies meeting the following criteria were included in this review: (a) published studies; (b) published in English; (c) randomized controlled trials (RCTs); (d) adult patients (age ≥ 18 years old) of both sexes who suffered or had suffered from any type of cancer at the time of the study; (e) patients in the process of receiving or who received an HSCT;(f) patients who had undergone an exercise program intervention; (g) studies with an effect on psycho-emotional and QoL factors in adult patients.

 Studies that were excluded in this review: (a) unpublished clinical trials registered on clinicals.gov; (b) grey literature (e.g., reports, conference proceedings, doctoral theses); (c) published and unpublished systematic reviews or meta-analyses; (d) studies that only related exercise and drugs.

The study selection process was conducted via step 1 (title and abstract screening) following by step 2 (full-text screening). In step 2, studies were selected only studies scored ≥ 6 in the Physiotherapy Evidence Database (PEDro) scale.

L152- I believe the authors have not clearly described when studies were scored according to the PEDRO scale. Was this done in Step 1 (abstract screening) or Step 2 (full text screening)?

Thank you for your response. A sentence has been added under "2.4. Assessment of risk of bias" describing that studies were scored according to the PEDRO scale in step 2 (full text screening).

L150-Please remove "Studies were 179 excluded until 24 August, 22. " from lines 179-180.

Thank you for your comment. This sentence has been removed from the section "2.4. Risk of Bias Assessment".

L131-The authors state: "Moreover, this method was follow by numerous systematic reviews, and is a cut-point of high methodological quality, for example:" and then provide two studies by the same group of authors as examples...

Thank you for your comment. We have added 2 other references from cancer patients as examples from outside the authors of this studies:

  1. Nelson NL. Breast Cancer-Related Lymphedema and Resistance Exercise: A Systematic Review. J Strength Cond Res. 2016;30(9):2656-2665. doi:10.1519/JSC.0000000000001355
  2. Juvet LK, Thune I, Elvsaas IKØ, et al. The effect of exercise on fatigue and physical functioning in breast cancer patients during and after treatment and at 6 months follow-up: A meta-analysis. Breast. 2017;33:166-177. doi:10.1016/j.breast.2017.04.003
  3. Hidalgo B, Hall T, Bossert J, Dugeny A, Cagnie B, Pitance L. The efficacy of manual therapy and exercise for treating non-specific neck pain: A systematic review. J Back Musculoskelet Rehabil. 2018;30(6):1149-1169. doi:10.3233/BMR-169615.
  4. Morales-Rodriguez E, Pérez-Bilbao T, San Juan AF, Calvo JL. Effects of Exercise Programs on Physical Factors and Safety in Adult Patients with Cancer and Haematopoietic Stem Cell Transplantation: A Systematic Review. Int J Environ Res Public Health. 2022 Jan 24;19(3):1288. doi: 10.3390/ijerph19031288. PMID: 35162312; PMCID: PMC8834842.
  5. Morales JS, Valenzuela PL, Herrera-Olivares AM, Baño-Rodrigo A, Castillo-García A, Rincón-Castanedo C, Martín-Ruiz A, San-Juan AF, Fiuza-Luces C, Lucia A. Exercise Interventions and Cardiovascular Health in Childhood Cancer: A Meta-analysis. Int J Sports Med. 2020 Mar;41(3):141-153. doi: 10.1055/a-1073-8104. Epub 2020 Jan 14. PMID: 31935777.

Figure 1 (flow chart) continues to not make sense. The authors excluded 653 studies, and 35 full texts were screened, however the number of reports excluded with reasons exceed the 35 full texts that were screened... Please revise.

Thank you for your comment. We have repeated the literature search and have completed the correct and detailed data in the flowchart. In addition, a new clarifying paragraph has been added to paragraph 3.1. Study selection (L157-161).

Systematic search, a summary description of the indexed terms used in Mesh for Pubmed and web of sciencie is given in section 2.1. The complete search is more extensive, which has been repeated to describe and through which existing errors have been corrected.

(bone marrow transplant OR hematopoietic stem cell transplantation) AND (exercise OR physical activity OR physical exercise OR acute exercise aerobic exercise OR exercise training OR supervised exercise program OR exercise program OR endurance exercise OR exercise therapy OR exercise tolerance) AND (immune system OR inmune system phenomena  OR inmune system diseases  OR plant immunity OR fatigue OR muscle fatigue  OR fatigue syndrome  OR alert fatigue OR infection  OR infection control  OR infection control practitioners OR focal infection OR cardiovascular function  OR cardiovascular physiological phenomenal  OR cardiovascular physiology  OR neuromuscular function  OR functional mobility OR basic activities of daily living OR ADL  OR Oxigen consumption OR strength OR muscle strength  OR shear strength OR compressive strength  OR resistance training OR acceleration OR mobility  OR mobility limitation OR range of motion , articular OR stress OR stress physiological OR oxidative stress OR mental fatigue OR occupational stress  OR anxiety OR performance anxiety OR anxiety  disorders  OR mental health OR health OR physical health  OR neoplasms OR cancer survivors).

On the other hand, the web of science database allows you to discard articles through the search filters before reaching step 1 (title and abstract).

The authors are not consistent in the formatting of Pedro, PEDRO, PEDro, etc.

Thank you for your comment. We have reviewed and adjusted PEDro's formatting throughout the manuscript.

The authors do not describe in detail the associations or effects of exercise program on the outcomes of interest, neither do they provide information on specific factors (anxiety, depression, functional vs symptoms of QoL) that improve or deteriorate with exercise programs during cancer treatment. The authors do not adequately describe if there are any particular exercise programs (intensity, duration etc) that significantly improved psycho-emotional and QoL factors.

Respect this sentence: The authors do not adequately describe if there are any particular exercise programs (intensity, duration etc), to See section 6. Discussion: L517-523.

“A few of the paragraphs in the discussion provide lists organized by letters. I believe the authors should try to avoid this. Please consider revising the discussion accordingly”.The authors changed the paragraphs in the discussion proving lists organized by letters to lists organized by number. I believe the authors should avoid the use of lists in the dicussion to increase readability and flow.

Thank you for your comment We have revised the discussion and organised the information differently.

The authors do not clearly show what the current study adds to the existing literature when describing the previous SR on the same/similar topics.

Thank you for your comment. We have highlighted further the differences between our systematic review and the previous ones in section "4. Discussion"(L342-345).
